# Impact of Time to Treatment Initiation on Quality of Response in Patients with Acute Myeloid Leukemia Receiving Intensive Chemotherapy

**DOI:** 10.3390/cancers17122028

**Published:** 2025-06-18

**Authors:** Elisa Buzzatti, Giovangiacinto Paterno, Raffaele Palmieri, Fabiana Esposito, Lucia Cardillo, Kristian Taka, Lucrezia De Marchi, Marco Zomparelli, Kleda Zaganjori, Flavia Mallegni, Elisa Meddi, Federico Moretti, Ilaria Cerroni, Carmelo Gurnari, Luca Maurillo, Francesco Buccisano, Adriano Venditti, Maria Ilaria Del Principe

**Affiliations:** 1Hematology, Department of Biomedicine and Prevention, University Tor Vergata of Rome, 00133 Rome, Italy; buzzattielisa@gmail.com (E.B.); raffaele.f.palmieri@gmail.com (R.P.); fabiana.e91@gmail.com (F.E.); lu-pa95@outlook.it (L.C.); taka.kristian@gmail.com (K.T.); lucrezia.demarchi@ptvonline.it (L.D.M.); marco.zomparelli98@gmail.com (M.Z.); kledazaganjori@gmail.com (K.Z.); flaviamallegni@gmail.com (F.M.); meddi.elisa@gmail.com (E.M.); frc.moretti@gmail.com (F.M.); ilaria.cerroni09@gmail.com (I.C.); francesco.buccisano@uniroma2.it (F.B.); dlpmlr00@uniroma2.it (M.I.D.P.); 2UOSD Mieloproliferative, Policlinico Tor Vergata, 00133 Rome, Italy; gg.paterno88@gmail.com (G.P.); luca.maurillo@uniroma2.it (L.M.); 3Department of Translational Hematology and Oncology Research, Taussig Cancer Institute, Cleveland Clinic, Cleveland, 44114 OH, USA

**Keywords:** acute myeloid leukemia, intensive chemotherapy, time to treatment, measurable residual disease

## Abstract

Nowadays, assessing patient fitness for intensive therapy and awaiting genetic data for better patient stratification led physicians to consider delaying treatment initiation in acute myeloid leukemia (AML). While the impact of this time to treatment (TTT) on survival and disease outcomes has been debated for years, its effect on achieving measurable residual disease (MRD) negativity remains underexplored. In this retrospective study of 196 adult AML patients receiving intensive chemotherapy, TTT was not associated with differences in overall survival or MRD negativity. These findings support that a short delay in therapy may be clinically acceptable and allows for thorough diagnostic evaluation to guide personalized treatment.

## 1. Introduction

Acute myeloid leukemia (AML) is a life-threatening disease for which prompt treatment is advised. Despite intensive protocols, the 5-year overall survival (OS) in AML ranges from 40–50% in younger patients to <10% in elderly and secondary cases [1]. The timing of treatment becomes crucial, especially for patients with high-risk features such as hyperleukocytosis and coagulopathy, as these conditions are linked with a significantly increased risk of early mortality. Nevertheless, a delay in therapy is frequently required to address patient comorbidities, treat concurrent infections and assess the biological characteristics of AML. In particular, the introduction of less intensive therapies highlighted the need for a more in-depth assessment of the patients and their comorbidities to determine their ‘fitness’ for intensive treatment, a process that demands considerable time for specialized assessments and imaging studies. Prior research has investigated the impact of time to treatment (TTT) on survival outcomes. For instance, Röllig’s large cohort study found no significant association between delayed treatment and survival, irrespective of age [2]. However, a recent meta-analysis [3] conducted in 2023 on 13 studies totaling 14.946 patients revealed substantial heterogeneity in the impact of prolonged TTT on various clinical outcomes, which depended on patient characteristics, treatment regimens, and other concomitant factors, thus stressing the complexity of this issue. With the advent of novel targeted therapies for AML, including FLT3 inhibitors, a critical question arises: does the delay in treatment initiation, necessary for comprehensive genomic profiling and patient fitness evaluation, outweigh the potential benefits of personalized treatment approaches? The answer is far from being clear as evaluating treatment success requires consideration of multiple outcomes beyond achievement of complete remission (CR), which may not always reflect a durable response. In fact, along with cytogenetic and molecular features, measurable residual disease (MRD) has proven to be a powerful prognostic indicator in AML, critical for both risk stratification and treatment decision-making [4,5,6]. Currently, MRD’s role is increasingly recognized, and ongoing studies are exploring its incorporation as a primary endpoint alongside more commonly used outcome measures like OS [7]. Patients achieving MRD negativity are known to have improved outcomes [8]; therefore, the introduction of novel therapies should aim both at increasing CR rates and enhancing the quality of response to treatment by ensuring a reduced risk of relapse. Therefore, assessing how treatment delay affects MRD gains greater importance, moving beyond the traditional emphasis on CR and OS seen in prior research. Based on these considerations, we described the impact of TTT measured after two cycles of intensive chemotherapy on a retrospective cohort of 196 patients.

## 2. Materials and Methods

The study was conducted on adult patients consecutively diagnosed with AML and receiving intensive chemotherapy at the Hematology Unit of Policlinico Tor Vergata Hospital between February 2010 and July 2023. Clinical and laboratory data were extracted from an anonymized database. The study protocol was approved by the local Ethics Committee and conducted in accordance with the Declaration of Helsinki. The exclusion criteria comprised age below 18 years and treatment with non-intensive chemotherapy or best supportive care. OS was calculated from diagnosis to death, last follow-up, or loss to follow-up. TTT was defined as the interval from AML diagnosis to the start of intensive chemotherapy. For the analysis, TTT was categorized into three groups: <8 days, 8–14 days, and >14 days. Early deaths (ED), according to the literature, were classified as death within 30 days from treatment start. Patient data included age, sex, white blood cell count (WBC), date of diagnosis and risk stratification according to the 2017 European Leukemia Net (ELN) guidelines [9]. The 2017 ELN classification was employed as most patients were enrolled prior to the introduction of the 2022 classification, and risk stratification and subsequent treatment allocation were based on this earlier system. Intensive chemotherapy was administered to patients using various protocols. The “7 + 3” regimen constituted the most frequent approach, incorporating midostaurin after its Italian approval in 2018. Additionally, fludarabine-based regimens, mitoxantrone-based regimens, and CPX-351 were utilized. Treatment data included chemotherapy regimens, start date, disease evaluation after one and two cycles. CR definition as for ELN 2017 was: bone marrow blasts <5%; absence of circulating blasts and blasts with Auer rods; absence of extramedullary disease; absolute neutrophil count ≥1.0 × 10^9^/L; platelet count ≥100 × 10^9^/L. For AML without molecular markers, MRD was evaluated after two cycles according to ELN MRD recommendations using a 8-multiparametric flow cytometry (MFC) assay with a sensitivity of 10^−3^ to 10^−5^. We used leukemia-associated immunophenotype (LAIP) and different from normal aberrant immunophenotype approaches to evaluate MFC MRD.

In presence of a molecular marker MRD assessment was performed using real-time quantitative PCR with a sensitivity of 10^−4^ to 10^−6^ [10]. MRD negativity criteria were <0.035% of blasts by MFC [5], <200 copies/10^4^ ABL for *NPM1* [11], <10 copies/mL peripheral blood for *CBFB::MYH11* [12], and a > 3 log10 reduction in *RUNX1::RUNXT1* transcript copies from diagnosis [13]. Patients were not included in the statistical analysis for MRD negativity if they lacked a reliable molecular marker, a LAIP, or a dependable initial phenotypic signature.

A binary logistic regression model was employed to assess the impact of TTT. Model fit was evaluated using Nagelkerke R-squared and the Hosmer-Lemeshow goodness-of-fit test. Odds ratios (ORs) were calculated to quantify the association between TTT and the likelihood of achieving CR and MRD negativity. OS was estimated using the Kaplan-Meier method. Log-rank test was used to compare survival curves among the different TTT groups. Statistical significance was defined as a two-sided *p*-value less than 0.05 (SPSS v28).

## 3. Results

The study cohort comprised 196 AML patients aged 21 to 78 years (median 57 years) with a slight male preponderance (52.8%). Median WBC was 11.11 × 10^9^/L (range 0.01–324 × 10^9^/L). Of the 196 patients, 94.5% were classified according to the 2017 ELN criteria, while 5.6% (11 patients) were unclassifiable due to cytogenetic failure and the absence of molecularly defined characteristics. Among the classified patients, 22.4% (44) were favorable risk, 42.9% (84) intermediate risk and 29.1% (57) adverse risk. Patients were treated according to “7 + 3” regimen in 54.5% of cases, fludarabine-based regimen were used in 20% of cases, mitoxantrone-based regimen in another 20% and CPX-351 in 5.1%. At disease onset, cytoreduction with hydroxyurea was performed in 44.8% of patients (n = 88), whereas leukapheresis was necessary in only 10% (n = 20). The TTT for the entire cohort ranged from 4 to 126 days with a median of 11 days. Overall, patients were more likely to receive treatment within the range of 8–14 days (35.7%, n = 70), followed by <8 days (34.7%, n = 68), and >14 days (29.6%, n = 58) (Table 1). The reasons for delayed treatment initiation post-diagnosis were infection management in 62 (31.6%) cases (mostly pneumonia and sepsis), comorbidity screening in 80 (40.9%) cases, and the requirement for molecular investigations in 54 (27.5%) cases. In cases where treatment initiation occurred beyond 14 days, the major contributing factors were multiple concurrent antimicrobial therapies (81%, n = 47) and fitness assessment (19%, n = 11).

Across all TTT groups, most of patients exhibited intermediate risk, with this group comprising more than 40% of patients in each timeframe. The proportion of patients with favorable risk was highest in the 8–14 day TTT group (29%), slightly high in the <8 day group (21%), and lowest in the >14 day group (18%). Notably, the proportion of patients with unclassifiable risk exceeded 10% in the <8 day TTT group. Conversely, the proportion of patients with adverse risk exhibited a linear increase across the TTT groups, with the highest proportion observed in the >14 day TTT group (35%). Median follow-up was 605 days (range 7–1900). Median OS was 414 days. OS when stratified by ELN categories, was consistent with guideline-reported outcomes [14]. The 3-year OS rate was 34%, stratified by TTT of <8, 8–14, >14 days was 36.1%, 33.5%, 26.4%, respectively. Kaplan-Meier analysis revealed no statistically significant difference in OS among patients with varying TTT (*p* = 0.48) (Figure 1a). No difference was found even stratifying patients for age (≤60 years *p* = 0.94, >60 years *p* = 0.25) (Figure 1b,c).

ED rate was 4% (8/196), but given the low number of events, no statistical analysis was performed. The 8 patients were distributed as follows: two in the <8 days category, three in the 8–14 days, and three in the >14 days.

A 75.5% (148/196) CR rate was observed in the study cohort (Table 2). Excluding the ED, forty patients (20.4%) did not achieve CR.

Statistical analysis demonstrated a significantly higher CR rate in patients with treatment initiation within 8–14 days (*p* = 0.004) and within 7 days of diagnosis (*p* = 0.006) compared to those who started treatment after 14 days (Table 3).

MRD assessment was feasible in 140 (71.4%) of the 196 patients in the study cohort. Excluding the ED and the 40 patients who did not achieve CR, in 8 patients it was not possible to assess MRD since they did not have a suitable LAIP available or a molecular marker. The percentage of patients who achieved MRD negativity after two cycles of therapy was 35% (49 out of 140), while the remaining 65% (91 out of 140) had a positive MRD (Table 4). No statistically significant difference in MRD negativity rates was found comparing the three TTT groups (Table 5).

## 4. Discussion

Our study adds to the growing body of evidence suggesting that treatment delay does not significantly impact survival.

Prior studies investigating the effect of TTT have yielded heterogeneous results, potentially influenced by factors such as varying cohort characteristics and potential selection biases. Our results align with four prior studies conducted between 2009 and 2020 demonstrating that TTT did not significantly impact OS [2,15,16,17]. However, in contrast to two of these, which reported an age-dependent association between TTT and OS, with younger patients experiencing worse outcomes with longer TTT [18,19], our study found no such correlation. Another large study (55,985 patients) [20] from the National Cancer Database reported an association between a treatment delay of five days and poorer survival in patients <60 years, however the statistical difference was small, and those results may be confounded by selection bias, rather than reflecting a direct causal impact of treatment delay.

Due to the small number of ED observed in our cohort, we did not conduct a formal analysis of ED compared to TTT. Anyhow, in five previous studies TTT did not influence mortality rate [2,15,16,21,22].

Regarding CR, only two previous studies reported a correlation with delayed treatment initiation. This association was observed depending on age: Sekeres et al. found that in younger patients, CR decreased with TTT >5 days, while Juliusson et al. showed that longer TTT was significantly associated with a lower rate of CR in older patients [18,21].

In our analysis, patients initiating treatment within 14 days exhibited a higher CR rate. However, the presence of a greater number of patients with adverse prognostic profiles in the >14 days treatment initiation group suggests a need for further investigation to determine the independent effect of treatment initiation timing.

To the best of our knowledge, this is the first time that MRD assessment was evaluated in comparison to TTT. In our analysis MRD negativity was reached in 35% of the patients, with no difference depending on TTT. This is critical since the quality of response is becoming increasingly important in AML, especially for transplant allocation and decision-making [23,24]. Nowadays, MRD plays a role as a predictive biomarker of relapse risk, a prognostic tool and a surrogate endpoint for clinical trials, therefore an analysis of the impact of treatment delay should include MRD among the considered outcomes. Our findings suggest that, while factors such as treatment type and biological characteristics of the disease are known to have an impact on achieving MRD negativity, TTT does not appear to be a significant influencing factor. In light of this, a reasonable delay of less than two-three weeks in treatment initiation allows for a more detailed genetic profiling of the leukemia clone, enabling the use of targeted therapies potentially leading to a deeper response. A potential future research could be to conduct a larger-scale analysis of patients stratified by ELN risk groups to determine whether the impact of delayed treatment on achieving MRD negativity varies across these risk categories. This would help us understand whether it is truly worthwhile to delay treatment to obtain more detailed biological information.

Another pivotal aspect of modern AML management is the assessment of patient ‘fitness’ at diagnosis, essential for determining treatment eligibility and optimizing patient outcomes. While age was previously a primary determinant, current guidelines [25,26], emphasize a comprehensive assessment of comorbidities, including cardiac, pulmonary, renal, and hepatic function, along with age and concurrent infectious diseases. This evaluation, though time-consuming, leads to a more personalized treatment approach, ensuring that patients receive the most appropriate therapy while minimizing treatment-related toxicity. The finding that TTT does not significantly affect outcomes or quality of response supports a more deliberate approach to patient care and allows the development of individualized treatment plans according to risk factors based on genetic profile and comorbidities.

Within the framework of non-intensive strategies, a recent study evaluated the impact of TTT in patients receiving venetoclax-based therapy [27]. The results demonstrated that a 10-day delay in treatment initiation did not significantly impact OS. This finding is particularly relevant for unfit patients, as a comprehensive assessment of comorbidities, infections, and concomitant events often requires additional time. Even so, an important limitation of this study was the lack of MRD assessment correlated with TTT since even in unfit patients, the status of MRD negativity has been shown to correlate with improved survival [28].

The present study has several limitations inherent in its retrospective and single-center nature. The retrospective data collection may have resulted in incomplete or inconsistent data capture, potentially introducing information bias. Furthermore, the single-center design restricts the external validity of our findings. In fact, the specific characteristics of our patient cohort, coupled with institutional treatment protocols and resource availability, may not accurately reflect the broader population of acute leukemia patients or practices in diverse healthcare settings. Consequently, caution is warranted when extrapolating these results to other centers or patient demographics.

Another limitation of this study stems from its expansive data collection period, which spanned a transformative era in acute leukemia management. This timeframe coincided with significant advancements in diagnostic techniques—including the evolution of molecular and genetic profiling—and substantial shifts in intensive treatment protocols.

Consequently, what was initially a strong inclination towards earlier treatment initiation gradually yielded to evolving prescribing practices. These practices progressively placed a greater emphasis on delaying therapy to await comprehensive genetic profiling results, a practice that became increasingly feasible and prioritized due to improved diagnostic capabilities. This inherent temporal variability in both diagnostic approaches and therapeutic strategies represents a potential confounding factor that warrants careful consideration when interpreting the study’s findings.

A significant limitation in current research is the absence of a universally accepted cut-off point for TTT. This variability in defining clinically relevant delays makes it difficult to compare results across different studies and limits the ability to generalize findings. In the present study, the 8-day cut-off was selected based on our institution’s technical capacity to efficiently perform the necessary diagnostic investigations within this timeframe, however, assessments of comorbidities and identification and treatment of active infections may require more time. Indeed, every study has employed arbitrary cut-offs, such as the 0–5, 6–10, 11–15, and >15-day categories used in Röllig’s study [2], or the simple ≤4 versus >4-day categorization used in another study [29]. Currently, this heterogeneity in cut-off points represents a pivotal factor impacting the methodological rigor and interpretability of such studies.

In fact those cutoffs are frequently contingent upon specific center-based protocols and the varying capabilities of institutions to handle associated complications.

Future prospective studies are needed to further investigate the impact of treatment timing and refine AML treatment strategies. These studies should feature more homogeneous patient cohorts, standardized treatment protocols, larger populations within narrower timeframes, and consistent cutoff definitions for treatment delays.

To facilitate personalized medicine though, there is a pressing need to expedite cytogenetic and molecular analyses, allowing for quicker identification of patients eligible for targeted therapies ensuring, therefore, a prompt treatment initiation.

## 5. Conclusions

In conclusion, patients’ survival is certainly influenced by well-known prognostic factors such as ELN risk group and MRD levels rather than the timing of treatment initiation [14,15].

Our findings suggest that TTT within the first 14 days after diagnosis does not significantly impact the likelihood of achieving MRD negativity in patients treated with intensive chemotherapy and therefore it does not affect the quality of response.

## Figures and Tables

**Figure 1 cancers-17-02028-f001:**
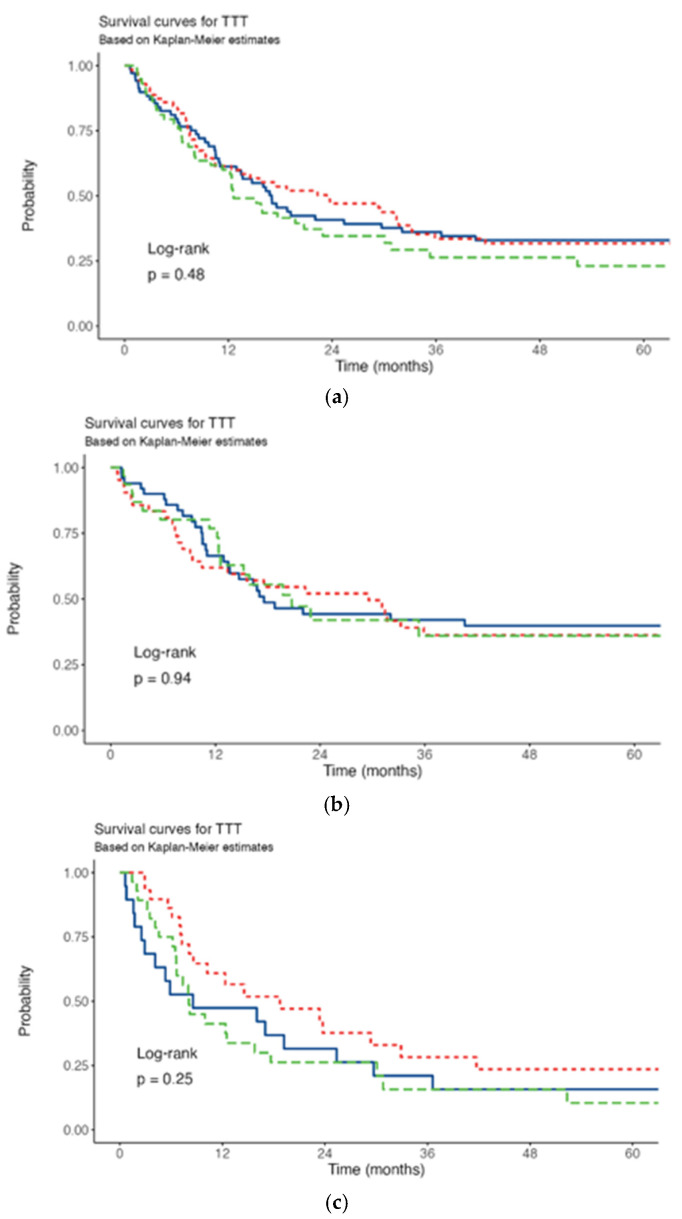
Survival curves. (**a**) All patients. (**b**) ≤60 years. (**c**) >60 years. Red line: TTT <8 days; blu line: TTT 8-14 days; green line: TTT >14 days.

**Table 1 cancers-17-02028-t001:** Patients’ characteristics in the three TTT groups.

	All n = 196	TTT <8 Days n = 68 (34.7%)	TTT 8–14 Days n = 70 (35.7%)	TTT >14 Days n = 58 (29.6%)
**Sex**				
Male (%)	104 (52.8)	32 (47)	40 (57.1)	32 (55.1)
Female (%)	92 (47.2)	36 (53)	30 (42.9)	26 (44.9)
**WBC median, (range) × 10^9^/L**	11.11 (0.01–324)	11.67 (0.21–200)	11.11 (0.37–324)	7.71 (0.01–302)
**Age median (range)**	57 (21–78)	54 (24–75)	58 (21–73)	59 (23–78)
**ELN risk group**				
Favorable n (%)	44 (22.4)	14 (21)	19 (29)	11 (18)
Intermediate n (%)	84 (42.9)	29 (43)	29 (40)	26 (45)
Adverse n (%)	57 (29.1)	16 (23)	21 (30)	20 (35)
Unclassifiable n (%)	11 (5.6)	9 (13)	1 (1)	1 (2)
**Treatment regimen**				
7 + 3 (%)	107 (54.5)	40 (58.8)	35 (50)	32 (55)
Fludarabine-based (%)	39 (20)	11 (16.2)	18 (25.7)	10 (17.2)
Mitoxantrone-based (%)	40 (20.4)	14 (20.5)	16 (22.9)	10 (17.2)
CPX-351 (%)	10 (5.1)	3 (4.5)	1 (1.4)	6 (10.6)

TTT: time to treatment; WBCc: white blood cell count; ELN: European Leukemia Net.

**Table 2 cancers-17-02028-t002:** Treatment outcomes CR, ED and 3-year OS of all patients stratified for TTT groups.

	All n = 196	TTT <8 Days n = 68	TTT 8–14 Days n = 70	TTT >14 Days n = 58
CR (%)	148 (75.5)	55 (80)	58 (83)	35 (60)
ED (%)	8 (4)	3 (4.4)	3 (4.2)	2 (3.4)
3-year OS (%) [95%CI]	34 [24.4–48]	36.1 [26.1–49.9]	33.5 [23.7–47.5]	26.4 [16.2–42.9]

TTT: time to treatment; CR: complete remission; ED: early deaths; OS: overall survival.

**Table 3 cancers-17-02028-t003:** Odds ratio for the achievement of CR.

	OR	95% CI	*p* Value
CR			
<8–>14 days	3.14	1.395–7.08	0.006
8–14–>14 days	3.31	1.474–7.45	0.004
<8–8–14 days	0.95	0.468–1867	0.89

OR: odds ratio; CI: confidence interval; CR: complete remission.

**Table 4 cancers-17-02028-t004:** MRD negativity distribution among the three TTT groups.

	All n = 140	TTT <8 Days n = 54	TTT 8–14 Days n = 54	TTT >14 Days n = 32
Negative MRD (%)	49 (35)	18 (33.3)	22 (40.7)	9 (28.1)

TTT: time to treatment; MRD: measurable residual disease.

**Table 5 cancers-17-02028-t005:** Odds ratio for the achievement of MRD negativity.

	OR	95% CI	*p* Value
Negative MRD			
<8–8–14 days	0.72	0.33–1.59	0.42
<8–>14 days	1.27	0.49–3.32	0.61
8–14–>14 days	0.81	0.39–1.69	0.49

OR: odds ratio; CI: confidence interval; MRD: measurable residual disease.

## Data Availability

All relevant data are included in this article. For additional data inquiries, a request can be directed to the corresponding author.

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
