# Peer review of "Impact of Time to Treatment Initiation on Quality of Response in Patients with Acute Myeloid Leukemia Receiving Intensive Chemotherapy"

_cancers, 2025, doi:10.3390/cancers17122028_

Round 1
Reviewer 1 Report
Comments and Suggestions for Authors
I read a manuscript entitled “Impact of Time to Treatment Initiation on Quality of Response 2 in Patients with Acute Myeloid Leukemia Receiving Intensive 3 chemotherapy” written by Dr. Elisa Buzzatti et al., which demonstrated how the time to treatment (TTT) influenced the clinical courses in the patients with acute myeloid leukemia (AML). This study brings a novel finding in AML therapy, however this retrospective study can include some bias, confounding, and interactions.
Major comments
- Do you advocate how long-time delay before the initiation of treatment can be acceptable?
- During median 11 days of TTT, how the patients were managed?
- You mentioned the time range of TTT category (till 7 days, till 14 days, and after 15 days) is “arbitral”. How would be the result possible if you analyzed the time category in shorter duration, for instance until 3 days and after.
- In developed countries, AML patients will be treated within 2 to 3 days after the diagnosis. How do you think this realistic practical?
- Please specify the genetic profile of the recruited patients. And much better, if analyzed according to the ELN risk category.
- Could you analyze the survival analysis according to the ELN risk stratification?
- The initial treatment (induction therapy) regimen were differed between the risk categories? If so, could it be possible to influence to the study result?
Minor comments
- Define the treatment protocol and regimen observed in the Method section.
- If the patients of delayed TTT group caught the infections more frequently at the onset, what were the infectious diseases?
- You mentioned the reasons of treatment delay was the time for infection treatment and screening result. Can those factors contribute to the survival analysis?
Author Response
I read a manuscript entitled “Impact of Time to Treatment Initiation on Quality of Response 2 in Patients with Acute Myeloid Leukemia Receiving Intensive 3 chemotherapy” written by Dr. Elisa Buzzatti et al., which demonstrated how the time to treatment (TTT) influenced the clinical courses in the patients with acute myeloid leukemia (AML). This study brings a novel finding in AML therapy, however this retrospective study can include some bias, confounding, and interactions.
We are grateful to the reviewer for the comments and for bringing attention to the potential bias of our analysis. Please find enclosed our rebuttal on a point-by-point basis. All changes introduced into the manuscript are recognizable in red. All the removed parts are identified with a strikethrough type.
Major comments
- Do you advocate how long-time delay before the initiation of treatment can be acceptable?
Thank you for the question. As stated in the manuscript, genetic assessment, comorbidities, and infection management may influence the delay to diagnosis; however, a 2-week delay doesn't appear to affect the achievement of MRD negativity.
- During median 11 days of TTT, how the patients were managed?
Thank you for the observation, we used hydroxyurea and/or leukapheresis during the waiting period prior to treatment as mentioned in lines 119-121.
- You mentioned the time range of TTT category (till 7 days, till 14 days, and after 15 days) is “arbitral”. How would be the result possible if you analyzed the time category in shorter duration, for instance until 3 days and after.
Thank you for the comment. We opted not to limit the therapeutic delay to just three days because a thorough patient evaluation seems challenging to complete within such a brief period. Indeed, none of our patients began intensive chemotherapy before the three-day mark.
- In developed countries, AML patients will be treated within 2 to 3 days after the diagnosis. How do you think this realistic practical?
We are very grateful for this insightful comment. We believe it is not realistic for the majority of patients to initiate treatment within three days. Even with the possibility of rapid genetic evaluation (turnaround time for cytogenetic analysis of 7-15 days, 2.5 days for FISH, 3-5 days for molecular testing*), the clinical and instrumental assessment of comorbidities and infection management would, on average, require more time.
* Voso MT, Ferrara F, Galimberti S, Rambaldi A, Venditti A. Diagnostic Workup of Acute Myeloid Leukemia: What Is Really Necessary? An Italian Survey. Front Oncol. 2022;12:828072. Published 2022 Feb 17. doi:10.3389/fonc.2022.828072
- Please specify the genetic profile of the recruited patients. And much better, if analyzed according to the ELN risk category.
Thank you for pointing that out. In line 113, we indeed detailed the categorization of patients according to the ELN 2017 guidelines, and Table 1 then illustrates the treatment delay observed within each of those risk categories.
- Could you analyze the survival analysis according to the ELN risk stratification?
We're very grateful for your comment.
We found that the overall survival, when stratified by ELN categories, was comparable to what's reported in AML treatment guidelines. For this reason, we felt it wasn't particularly informative to include this specific stratification in our work. Instead, we chose to focus on the impact of treatment delay, which we believed offered a more novel and relevant insight for our study. However, we added a sentence in the manuscript following your comment.
- The initial treatment (induction therapy) regimen were differed between the risk categories? If so, could it be possible to influence to the study result?
We agree with the reviewer that treatment regimens naturally vary across risk categories, as genetic profiles directly guide therapeutic choices. In our limitations in fact, we highlighted that the heterogeneity of treatments, spanning several years, may have influenced the analysis.
Minor comments
- Define the treatment protocol and regimen observed in the Method section.
Modified according to suggestion.
- If the patients of delayed TTT group caught the infections more frequently at the onset, what were the infectious diseases?
Thank you for your valuable comment. As indicated in line 128, 62 patients did experience infections requiring treatment. Following your suggestion, I've now specified the types of infections within the manuscript for greater clarity.
- You mentioned the reasons of treatment delay was the time for infection treatment and screening result. Can those factors contribute to the survival analysis?
In our work, these evaluations/treatments, which resulted in a delay to treatment, did not impact survival, as stated in lines 145-146.
Reviewer 2 Report
Comments and Suggestions for Authors
Thank you for the opportunity to review this manuscript, which addresses a timely and clinically relevant question: the impact of time to treatment initiation on both overall survival and the achievement of measurable residual disease negativity in patients with acute myeloid leukemia receiving intensive chemotherapy. The retrospective design and incorporation of MRD as an endpoint are valuable; however, the manuscript in its current form requires substantial revision.
First, the abstract needs revision for clarity. The abstract does not fully convey the significance of MRD as a modern surrogate marker in AML or provide sufficient context for the rationale behind the selected TTT intervals.
In the introduction, a more compelling justification is needed to explain why assessing MRD in this context adds value beyond prior studies that focused solely on CR or OS. The introduction should reflect this shift more clearly.
The methods section, though generally complete, lacks detail in several areas. The criteria for defining TTT cutoffs appear institution-specific and could limit the generalizability of the findings. The authors should provide further rationale, particularly as different studies use various thresholds. The MRD evaluation methods are only briefly described. It is important to specify the assay sensitivity, consistency in application over time, and how variations in flow cytometry versus PCR techniques were handled, especially given the study period spans over a decade. There is also a lack of information on how missing MRD data were managed statistically.
The limitations of the study are only partially addressed. The long study period, spanning major changes in AML diagnostics and therapy, is a potential source of confounding that must be more deeply explored. The single-center nature of the study further restricts its external validity. In addition, the heterogeneity in TTT cutoffs used in different studies remains a major barrier in comparing outcomes, and the authors should expand their discussion of how institutional practices influence the perceived impact of TTT.
Sincerely.
Comments on the Quality of English LanguageFinally, the manuscript is affected by multiple formatting inconsistencies and typographical errors.
Author Response
Thank you for the opportunity to review this manuscript, which addresses a timely and clinically relevant question: the impact of time to treatment initiation on both overall survival and the achievement of measurable residual disease negativity in patients with acute myeloid leukemia receiving intensive chemotherapy. The retrospective design and incorporation of MRD as an endpoint are valuable; however, the manuscript in its current form requires substantial revision.
We are grateful to the reviewer for the comments. Please find enclosed our rebuttal on a point-by-point basis. All changes introduced into the manuscript are recognizable in red. All the removed parts are identified with a strikethrough type.
First, the abstract needs revision for clarity. The abstract does not fully convey the significance of MRD as a modern surrogate marker in AML or provide sufficient context for the rationale behind the selected TTT intervals.
Thank you for you comment. We modified the abstract according to your suggestions.
In the introduction, a more compelling justification is needed to explain why assessing MRD in this context adds value beyond prior studies that focused solely on CR or OS. The introduction should reflect this shift more clearly.
Thanks for bringing this to our attention. The role of MRD is described from line 68 to line 80, but we've added a sentence to further emphasize the importance of MRD assessment in the context of treatment delays in these patients.
The methods section, though generally complete, lacks detail in several areas. The criteria for defining TTT cutoffs appear institution-specific and could limit the generalizability of the findings. The authors should provide further rationale, particularly as different studies use various thresholds. The MRD evaluation methods are only briefly described. It is important to specify the assay sensitivity, consistency in application over time, and how variations in flow cytometry versus PCR techniques were handled, especially given the study period spans over a decade. There is also a lack of information on how missing MRD data were managed statistically.
We are very grateful for this comment and we decided to expand the methods section according to your suggestions.
The limitations of the study are only partially addressed. The long study period, spanning major changes in AML diagnostics and therapy, is a potential source of confounding that must be more deeply explored. The single-center nature of the study further restricts its external validity. In addition, the heterogeneity in TTT cutoffs used in different studies remains a major barrier in comparing outcomes, and the authors should expand their discussion of how institutional practices influence the perceived impact of TTT.
Thank you for your insightful comment; it provided a valuable opportunity to enhance our discussion, particularly regarding the study's limitations. We agree that several key limitations warrant further consideration.
Round 2
Reviewer 1 Report
Comments and Suggestions for Authors
The manuscript was promptly revised.
Reviewer 2 Report
Comments and Suggestions for Authors
No comments